# A Study of the Gelatin Low-Temperature Deposition Manufacturing Forming Process Based on Fluid Numerical Simulation

**DOI:** 10.3390/foods12142687

**Published:** 2023-07-12

**Authors:** Qiang Tong, Wentao Zhao, Tairong Guo, Dequan Wang, Xiuping Dong

**Affiliations:** 1College of Mechanical Engineering and Automation, Dalian Polytechnic University, Dalian 116034, China; tongqiang.work@outlook.com (Q.T.); zwt2363038548@outlook.com (W.Z.); gtr446962334@outlook.com (T.G.); work.ty1122@outlook.com (D.W.); 2School of Food Science and Technology, Dalian Polytechnic University, Dalian 116034, China

**Keywords:** low-temperature forming, numerical simulation, rheological properties, printing temperature

## Abstract

Low-temperature deposition manufacturing has attracted much attention as a novel printing method, bringing new opportunities and directions for the development of biological 3D printing and complex-shaped food printing. In this article, we investigated the rheological and printing properties of gelatin solution and conducted numerical simulation and experimental research on the low-temperature extrusion process of gelatin solution. The velocity, local shear rate, viscosity, and pressure distribution of the material in the extrusion process were calculated using Comsol software. The effects of the initial temperature, inlet velocity, and print head diameter of the material on the flow field distribution and printing quality were explored. The results show that: (1) the fluidity and mechanical properties of gelatin solution vary with its concentration; (2) the initial temperature of material, inlet velocity, and print head diameter all have varying degrees of influence on the distribution of the flow field; (3) the concentration change of the material mainly affects the pressure distribution in the flow channel; (4) the greater the inlet velocity, the greater the velocity and shear rate in the flow field and the higher the temperature of the material in the outlet section; and (5) the higher the initial temperature of the gel, the lower the viscosity in the flow field. This article is of great reference value for the low-temperature 3D printing of colloidal materials that are difficult to form at room temperature.

## 1. Introduction

3D printing technology refers to a technique employed for creating entities with special shapes/properties by stacking materials layer by layer based on digital models, and is capable of completing multiple tedious processes in traditional machining processes with a high degree of flexibility and intelligence in the absence of a mold as an alternative to traditional processing lines, thereby reducing energy consumption and production cost [1]. There are three categories of raw materials commonly used for 3D printing, namely powder, slurry, and solid. The commonly used printing methods for powder raw materials include 3D printing technology (3DP) and selective laser sintering technology (SLS), those for solid raw materials include inkjet-forming technology (LJP) and digital light technology (SLA), and those for slurry raw materials mainly include fused deposition technology (FDM), direct-write printing technology (DIW) [2], and low-temperature deposition manufacturing (LDM). LDM is available as either plunger or screw types, depending on how extrusion pressure is provided. The gel material itself has a certain viscosity. Screw printing requires direct contact between the screw and material, making it difficult to clean during actual printing [3,4]. However, plunger printing is simple in structure without requiring contact with the material, and the stepper motor is directly connected to a piston with high accuracy [5].

Gels typically consist of large amounts of water and a three-dimensional network of cross-linked polymers. Multifunctional hydrogel inks include double-network hydrogels, magnetic hydrogels, temperature-sensitive hydrogels, and biogels [6]. Gelatin (GEL) is a typical temperature-sensitive gel, which is the product of animal collagen decomposition. It is an immune reaction-free and non-toxic natural polymer material with good biocompatibility in living organisms. It has been approved by the FDA to be widely used in the food industry as well as in biomedical fields such as pharmaceuticals, drug delivery, and tissue engineering [7]. Currently, in the food field, Yang et al. [8] have achieved chicken gelatin gel printing, and Dong et al. [9] have achieved potato starch gelatin gel printing, laying a theoretical foundation for digitally accurate design and regulation of food nutrition; in the biological field, Schuurman et al. [10] mixed methacrylate-anhydride gelatin solution with cartilage cells to prepare a multilayer lattice-structured bio-scaffold by 3D printing technology. However, all the above studies were performed from the perspective of material modification. Due to the limitations of the traditional preparation process, only hydrogels with two-dimensional or simple three-dimensional structures can be prepared, and those with high-precision complex models cannot be printed. In addition, the biological activity of materials cannot be perfectly guaranteed by the vast majority of printing technologies. Cryogenic 3D printing technology, from a process perspective, controls the temperature of the printing environment, and is not only able to print some hydrogels that are difficult to form but also effectively prevents the thermal phase transition of the material in the printing process while ensuring its biological activity [11]. Therefore, low-temperature 3D printing technology is widely used for the rapid design and manufacturing of hydrogels. Tan et al. [6] conducted a preliminary biological evaluation of 3D printing materials coated with type I collagen, poly-L-lysine and gelatin, and therefore, this article investigates the gelatin printing and forming process based on low-temperature deposition 3D printing technology from a process perspective, which has important research value.

Through the research results of Gu et al., Levett et al., and Zhao [12,13,14,15], it was found that when the concentration of gelatin aqueous solution is too high, it is not suitable for cell survival and is relatively easy to form. The research has little significance. When the concentration of gelatin aqueous solution is low, it is suitable for cell survival but is not easy to print and shape. In the food industry, gelatin is a natural food additive with a small amount of addition. Therefore, we chose gelatin aqueous solutions with concentrations of 12 wt%, 9 wt%, 6 wt%, and 3 wt% within this range to study their molding process.

The solutions of non-Newtonian fluid and nonlinear flow problems based on the finite element method (FEM) have been widely used in the additive manufacturability of material flow characteristics and is of great significance to 3D printing process optimization. The location of the printing head in a closed environment makes it very difficult to observe the changes in fluid temperature and characteristics in the 3D printing process [16].

Moreover, gelatin is sensitive to temperature; its viscosity and elasticity are sensitive to temperature changes. The ink near the nozzle outlet easily solidifies [17], resulting in nozzle blockage. Therefore, in this article we numerically simulated flow field distribution (speed, shear rate, viscosity) and temperature field distribution (temperature distribution, and the change of outlet material temperature over time) in the print head during printing using Comsol software. The mathematical model between extrusion line diameter and layer thickness provided a theoretical reference for the matching of inlet velocity and printing speed, and a self-made low-temperature 3D printing device was used in the gelatin printing process experiment, which provided a theoretical basis for the 3D printing of colloidal materials that are difficult to form at room temperature.

## 2. Materials and Methods

### 2.1. Materials

12 g, 9 g, 6 g, and 3 g of gelatin (purchased from Henan Sugar Cabinet Co., Ltd, Shangqiu, China. using food grade 180 freezing force) were, respectively, weighed and mixed with 88 g, 91 g, 94 g, and 97 g of water in a beaker to prepare 12 wt%, 9 wt%, 6 wt%, and 3 wt% gelatin aqueous solution. Subsequently, the gelatin solutions were placed in a constant temperature water bath heating pot (purchased by 600 W Qun’an Experimental Instrument Co., Ltd., Huzhou, China) for melting at a constant temperature of 40 °C, and then in a negative pressure box (Puruiqi AP-01P) for 3–5 min for defoaming treatment in an environment of −2.5 MPa. Finally, they were separately stored at 35 °C, 30 °C, and 25 °C for further use.

### 2.2. Measurement of Rheological Properties

The rheological properties of the samples were measured using the hybrid rheometer Discovery HR-1, and the viscosity parameters of gelatin solutions at 35 °C, 30 °C, and 25 °C were measured. Three parallel experiments were conducted on each group of samples. During the measurement, gelatin solution was dropped onto a parallel plate of the rheometer with a gap of 1 mm between the two parallel plates. Before the experiment, excess material outside the plate was scraped out and allowed to stand for 2 min until it became steady. When the temperature of the material was consistent with the preset temperature, the viscosity parameters were measured at a shear rate of 0.1–100 s^−1^ [5]. In the temperature-sensitive processes, the oscillation mode was used to test the viscosity variation curve with the temperature at 0–45 °C [18].

### 2.3. Finite Element Simulation

#### 2.3.1. Simulation Scheme

Material concentration, inlet velocity, and nozzle diameter are important parameters that affect flow field distribution in the flow channel. Gelatin is sensitive to temperature, so temperature distribution and changes in the printing process are also key factors affecting printing quality. In this article, we calculated the velocity, shear rate, viscosity, and pressure distribution of materials in the extrusion process by combining the coupling experiment and simulation while controlling a single variable, and then explored the effects of different process parameters (material concentration, inlet velocity, nozzle diameter, initial temperature of gelatin solution) on the flow field distribution and printing quality in the flow channel.

#### 2.3.2. Conditions and Assumptions

Assuming that the material was an incompressible fluid;Assuming that there was no slip between the material and the cylinder wall during extrusion;The piston moved at a constant speed at the inlet. The equivalent inlet velocity boundary conditions were applied on this boundary to replace the constant movement of the piston, with wall normal velocity v_n_ = 0 and surface velocity v_s_ = 0;The material was a fluid with high viscosity, whose inertial force and gravity were much smaller than the viscous force, so gravity was ignored;The pressure at the nozzle outlet was 101,325 Pa, which was 1 atm;The material was fully filled in the mold and stably flowed laminar in the flow channel.

#### 2.3.3. Fluid Viscosity Model

The viscosity of the gelatin solution decreased with the increase in shear rate, characteristic of a pseudoplastic fluid, i.e., the shear stress of the fluid τ and shear rate *γ* have a nonlinear correlation. The selection of appropriate fluid viscosity model is very important for studying the fluidity of a pseudoplastic fluid. The constitutive equation of a pseudoplastic fluid is [19]:(1)$$\eta_{a}=\tau /\gamma =k\gamma^{n-1}$$
where *k* is the consistency coefficient and *n* is the rheological index.

#### 2.3.4. Physical Model

Figure 1a shows a three-dimensional physical model of an extrusion device for a 3D printer, consisting of a material barrel, printing nozzle, fixed block, silicone heating sleeve, resistance wire, and materials. The outer diameter of the material barrel was 17 mm, the inner diameter was 14 mm, the length from inlet to outlet was 58 mm, the diameter of the print head outlet was 1 mm, the length of the fixed block was 45 mm, and the width was 30 mm, as shown in Figure 1b, which depicts the grid diagram of the extrusion device, which was automatically generated based on the physical field, with a total of 9,893,297 units.

##### Model Heat Transfer Analysis

The extrusion process of the gelatin solutions involved non-isothermal flow and solid heat transfer, and transient analysis of the extrusion process was conducted using Comsol software coupled with multiple physical fields. The model involved natural convection heat transfer and contact heat transfer, where natural convection heat transfer refers to the heat exchange between the air and the material in the material storage barrel, as well as that between the fixed block, the bottom of the material barrel, and the surface of the printing head and the external air, and contact heat exchange refers to the heat exchange between the material and the barrel wall, as well as that between the fixed block and the heating sleeve and the barrel wall.

##### The Heat Source and Its Initial Conditions

A silicone heating strip with a power of 20 W, a height of 45 mm, and a length of 106 mm was used as the heat source. A thermocouple was placed between the silicone heating strip and the outer wall of the material barrel. When the ambient temperature was 5 °C and the heating temperature of the silicone heating sleeve was 45 °C, the actual measured temperature of the outer wall of the storage barrel was 31 °C, the temperature of the inner wall of the storage barrel was 24.3 °C, and the temperature of the fixed block was 13.5 °C. Therefore, the initial temperature of the storage barrel and the printing head was set to 24.3 °C, and that of the fixed block to 13.5 °C.

##### Heating Condition Setting

The actual measured temperature of the outer wall of the storage barrel was 31 °C. The temperature of the thermocouple was used to replace the temperature of the outer wall of the material barrel. Using the implicit event built into Comsol, b1 = aveop1 (T) − 32 [degC] and b2 = aveop1 (T) − 30 [degC] were set. During the calculation: if b_1 > 0, stop heating; If b_2 < 0, start heating, where aveop1 (T) is the average temperature at the thermocouple, keeping the outer wall temperature of the material barrel constant at 31 °C.

#### 2.3.5. Convective Heat Transfer Coefficient

The heat transfer in the model was mainly in the form of natural convection heat transfer, and the empirical formula for the convective heat transfer coefficient of natural convection is [20].
(2)$$h_{c}=\frac{N_{\mu }}{L}$$

In formula (2), *L* is the characteristic length, *hc* is the Nusselt number, and $N_{\mu }$ is a dimensionless number that represents the intensity of convective heat transfer, which can be calculated by:(3)$$N_{\mu }=c{\left(G_{r}P_{r}\right)}^{m}$$

The material barrel was vertically placed, and the material was in the laminar flow state, so *c* was taken as 0.59, and *m* as 0.25.
(4)$$G_{r}=\frac{\beta g\Delta T\rho^{2}L^{3}}{\mu^{2}}$$
(5)$$P_{T}=\frac{c_{p}\mu }{k}$$
where $G_{r}$ is the Grashof number, which is dimensionless and represents the ratio of buoyancy and viscous force acting on the fluid;$\;P_{T}$ is the Prandtl number, indicating the relationship between the temperature boundary layer and the flow boundary layer; β is the coefficient of volume change; g is the acceleration of gravity; ΔT is the temperature difference; ρ is the density of air; L is the feature length; μ is the aerodynamic viscosity; $c_{p}$ is the specific heat capacity of air; and k is the thermal conductivity of the air.

#### 2.3.6. Printing Test

A self-developed LDM (Low-Temperature Deposition Manufacturing) printer was used for process experiments, as displayed in Figure 2a. The equipment diagram is shown in Figure 2b. The piston extrusion, printing environment cooling (5–10 °C), and heating of the silicone heating sleeve (35–55 °C) were utilized, with a printing speed of 2–8 mm/s and an inlet velocity of 0.03–0.09 mm/s. The printing model is shown in Figure 3.

## 3. Results and Discussion

### 3.1. Rheological Properties

Rheological performance is an important indicator for evaluating the 3D printing performance of materials [21]. Materials used for extrusion 3D printing must have certain flowability or melting characteristics, as well as the ability to maintain product shape after printing is completed [22]. As shown in Figure 4a, when shear rate fluctuated from 0.1 s^−1^ to 3 s^−1^, the apparent viscosity of the gelatin solution significantly decreased; when shear rate continued to increase from 3 s^−1^ to 100 s^−1^, the apparent viscosity of the gelatin solution decreased slowly. The curve tended to be flat as the viscosity approached 0 Pa·s, and the gelatin solution became closer to a Newtonian fluid. Overall, the apparent viscosity decreased with the increase in shear rate, indicating that the gelatin solution exhibited shear thinning behavior and was characteristically a pseudoplastic fluid, which was conducive to the discharge of slurry through the nozzle [23]. As shown in Figure 4c,d, the viscosity of the gelatin solution followed the same trend with shear rate at different temperatures, with higher temperatures resulting in lower viscosity and shear stress at the same shear rate. As shown in Figure 4a,b, the higher the concentration of gelatin solution, the higher the viscosity and the higher the shear stress at the same shear rate.

Viscosity measurement is one of the commonly used methods to determine the gel point temperature. The viscosity mutation point was used to determine the gel point temperature. As shown in Figure 4e, the curve of 9% gelatin viscosity changes with temperature. The gelatin viscosity changes abruptly at 31 °C, so 31 °C is the gel point temperature of 9% gelatin.

The storage modulus (G′) is a measure of elastic solid materials, which reflects the mechanical strength of the specimen. Materials with high mechanical strength exhibit excellent self-supporting properties after deposition and maintain a stable shape after printing. The loss modulus (G″) is the viscous response of the stress–strain ratio under dynamic oscillation frequency [23]. Different viscoelastic behaviors were illustrated using loss tangent (Tan δ = G′/G″) as a characteristic parameter. A tan δ less than 1 indicates mainly elastic properties, exhibiting solid characteristics. On the contrary, a tan δ greater than 1 indicates liquid properties [24], as shown in Figure 4f. When the temperature was greater than 31 °C, the gelatin solution tan δ was >1, and exhibited liquid properties; at a temperature less than 31 °C the gelatin solution tan δ was <1, and exhibited elastic properties.

### 3.2. Analysis of the Flow Field Distribution in the Flow Channel

Taking a gelatin aqueous solution with an inlet velocity of 0.03 mm/s, an initial temperature of 30 °C, a heating jacket temperature of 45 °C, an ambient temperature of 5 °C, and a concentration of 9 wt% as an example, the flow field distribution in the flow channel was analyzed.

#### 3.2.1. Temperature Distribution

The temperature distribution during the printing process was under the above process conditions as shown in Figure 5a. The heat exchange between the gelatin solution and the air in the barrel and the barrel wall resulted in the temperature gradient in the entire model. The temperature at the bottom of the material barrel was about 23 °C, higher than the surface temperature of the print head (about 20 °C), forming a small temperature gradient, which was conducive to the rapid cooling and solidification of materials after extrusion.

As shown in Figure 5b, the temperature at middle of the barrel was the highest, while the temperature at the fixed block was the lowest. Due to the influence of the fixed block temperature, the temperature of the gelatin solution at the inlet was relatively low.

#### 3.2.2. Velocity Distribution

The velocity field has a direct impact on the quality of 3D printing of materials. Figure 5c and Figure 5d, respectively, show the velocity field distribution in the flow channel and outlet under the above process conditions. It can be intuitively seen from Figure 5c that the material in the cylinder was in a low-speed state, with an average speed of 0.032 mm/s and that there was no significant change in speed. The material speed in the printing nozzle was relatively high, with an average of 1.82 mm/s. As can be seen from Figure 5d, on the same horizontal section, the fluid velocity peaked at the central axis of the nozzle and decreased from the central axis to the inner wall, with the lowest velocity near the inner wall, which was caused by the friction between the material and the solid wall [25]. These results are comparable to those found in the study by Liu et al. [26].

#### 3.2.3. Shear Rate Distribution

Gelatin solution is a pseudoplastic fluid whose viscosity varies with shear rate. Figure 5e and Figure 5f, respectively, show the distribution of shear rate fields in the flow channel and outlet section under the above process conditions. As shown in Figure 5e, the shear rate in the storage barrel section was lower, while that in the nozzle section was higher. As shown in Figure 5f, the shear rate in the flow channel bottomed at the central axis and increased from the central axis to the inner wall, with the shear rate higher near the inner wall, indicating that the material was subjected to significant shear force at the wall surface. These results are comparable to those of the study by Yang et al. [5].

#### 3.2.4. Viscosity Distribution

The viscosity distribution of materials is also one of the main factors affecting the forming quality of 3D printing, which is mainly reflected in the difficulty of extruding materials at the outlet. Gelatin is a thermosensitive material. When the temperature was lower than 30 °C, viscosity changed significantly with temperature. Figure 5g,h show the viscosity field distribution at the flow channel and outlet under the above process conditions, respectively. It can be intuitively seen from Figure 5g that the viscosity at the upper end of the barrel was high, and that at the center and at the bottom of material barrel was low. The viscosity at both sides was high, which was consistent with the temperature distribution at the same inlet velocity rate, indicating that temperature was one of the main factors affecting the viscosity distribution during printing.

As shown in Figure 5h, on the same horizontal section, the viscosity of the fluid decreased from the axis to the inner wall, with the lower viscosity near the wall surface. Due to the high shear rate at the nozzle outlet, the viscosity of the material dropped, indicating that the material was easier to extrude. These results are comparable to those reported by Shamsudin et al. for the effect of temperature and shear rate on viscosity distribution [27].

#### 3.2.5. Pressure Distribution

In the continuous-extrusion 3D printing process, pressure has a significant impact on the material extrusion process [28] Figure 5i and Figure 5j, respectively, show the pressure field distribution in the flow channel and outlet section under the above process conditions. As shown in Figure 5i, pressure in the storage barrel slightly changed, while that in the nozzle changed significantly. In addition, the pressure in the material barrel section was relatively large, while that in the nozzle section was relatively small. As shown in Figure 5j, the pressure in the flow channel increased from the axis to the inner wall, with that near the inner wall being relatively high, which was consistent with the horizontal velocity distribution inside the nozzle, both caused by the high friction between the inner wall and the material. These results are comparable to those of the study by Yang et al. [5].

### 3.3. Effect of Inlet Velocity on Flow Field Distribution

In the LDM process, inlet velocity was regulated by varying the velocity of the piston. The change in inlet velocity directly affected the velocity field within the flow channel, which was a key factor affecting flow channel parameters and 3D printing efficiency. Therefore, this section investigates the effect of inlet velocity on the flow field distribution within the flow channel.

As shown in Figure 6a,b, inlet velocity increased from 0.03 mm/s to 0.09 mm/s, and the average velocity at the nozzle outlet increased from 33.2 mm/s to 107.4 mm/s. The average shear rate at the outlet increased from 30 s^−1^ to 87.8 s^−1^. The above results indicated that both velocity and shear rate increased with the increase in inlet velocity. This is comparable to the findings of Emin et al. [29].

As shown in Figure 6c, when the initial temperature was 30 °C, the temperature variation trend in the Z-axis at different inlet velocities was the same, which firstly increased and then decreased. As the inlet velocity increased, peak temperature gradually shifted from the middle part of the printing barrel to the printing nozzle.

As shown in Figure 6d, at the initial temperature of 30 °C, with the increase in time, the temperature of the gelatin solution outlet section first decreased and then shifted towards equilibrium, and the greater the inlet velocity, the slower the decrease in temperature of the gelatin solution in the outlet section, and the greater the temperature at equilibrium. The inlet velocity increased from 0.03 mm/s to 0.09 mm/s, corresponding to the increased temperature of the outlet gelatin solution from 25.1 °C to 27.1 °C, respectively.

Temperature was one of the main factors affecting the viscosity of gelatin, as shown in Figure 6e,f. The variation trend of viscosity in different inlet velocity channels was the same, with viscosity first remaining horizontal and then gradually decreasing to point A because the temperature from the inlet to point A (the highest temperature point) gradually increased due to the influence of the heating sleeve, and with the increase in temperature the viscosity of the gelatin solution decreased.

Both extrusion speed and nozzle movement speed affect 3D printing because they change the extrusion amount per unit time and length. The key problem of the continuous extrusion line in 3D printing is matching between the extrusion speed and printing speed (Figure 7). In the deposition process, the cross-section of deposited silk thread is generally taken as an elliptical flat section, which increases the contact area between the front and back deposition layers, thereby enhancing the adhesion between the two [30].

The inlet velocity of the gelatin solution is set as *v_j_*, the cross-sectional area during extrusion as *s_j_*, the deposition velocity of the gelatin solution as *v_c_*, the cross-sectional area during deposition as *s_c_*, and the volume of gelatin solution extruded within time *t* is *Q*. To avoid line breakage or accumulation, the extrusion volume of gelatin solution per unit time should be equal to the deposition volume of gelatin solution, which can be represented by Equation (6).
(6)$$Q=v_{j}s_{j}t=vc_{s}c_{t\;}$$

Equation (7) is an expansion of Equation (6).
(7)$$v_{j}\pi {\left(d_{1}/2\right)}^{2}=v_{c}\left((d_{2}-h\right)h+\pi \left(h/2{)}^{2}\right)$$

Equation (8) is the relationship between inlet speed and printing speed, and the *K* value is shown in Equation (9).
(8)$$v_{c}=kv_{j}$$
(9)$$k=\frac{\pi d_{1}^{2}}{4d_{2}h-4h^{2}+\pi h^{2}}$$
(10)$$d_{2}=\frac{v_{j}\pi d_{1}^{2}}{4v_{c}h}+\frac{\left(4-\pi \right)h}{4}$$
where *v_j_* is inlet velocity; *v_c_* is deposition rate; *d* is the print head diameter; *d*_1_ is the inner diameter of the storage barrel; *d*_2_ is the width of extrusion line; and h is the thickness of deposition layer. Due to certain errors between the theoretical model and the actual situation, it was necessary to make corrections based on actual formula experiments.

Actual parameters were: *h* = 0.3 mm, *d* = 1 mm, *d*_1_ = 14 mm, *v_c_* = 5 mm/s. The theoretical widths of deposition lines corresponding to inlet velocities of 0.0156 mm/s, 0.0313 mm/s, 0.0469 mm/s, and 0.0625 mm/s were calculated by formula 10 to be 1.068 mm, 2.034 mm, 2.99 mm, and 3.955 mm, respectively. The actual widths of the deposition lines were measured using an electron microscope (TD-4KHT, SANQID), as shown in Figure 8, which were 1.660 mm, 2.280 mm, 2.830 mm, and 3.320 mm, respectively.

The relationship between the actual line width and the entrance velocity according to the linear fit of the Origin software is shown in Equation (11).
(11)$$d_{3}=1.1415+35.3344v_{j}$$

The theoretical width as a function of entrance velocity is shown in Equation (12).
(12)$$d_{2}=0.1075+61.529v_{j}\;$$

The width error between the actual width and the theoretical width is shown in Equation (13).
(13)$$Dd_{3}-d_{2}=1.0345-26.195v_{j}$$

The actual width minus the width error D is equal to the theoretical width, and the revised coefficient *k*_1_ can be obtained by bringing it into Formula (10). Formula (14) is the revised velocity relationship equation.
(14)$$v_{c}=k_{1}v_{j}$$
(15)$$k_{1}=\frac{\pi d_{1}^{2}}{4hd_{2}+\left(\pi -4\right)h^{2}-4.14h+104.78hv_{j}}$$

The appropriate line width was selected based on the printed model, with a wall thickness of 2 mm. Therefore, *d*_2_ was set to 2 mm. According to Formula (14), the optimal inlet velocity should be 0.0242 mm/s when the line diameter is 2 mm and printing speed is 5 mm/s. Using this speed as a reference, the effect of inlet velocity on 3D printing was studied by controlling a single experimental variable. Grid model printing experiments were conducted with inlet velocities of 0.0156 mm/s, 0.0250 mm/s, 0.0343 mm/s, and 0.0438 mm/s at a printing speed of 5 mm/s, respectively.

As shown in Figure 9a, there was stacking in the printing model at the inlet velocity of 0.0438 mm/s, indicating that inlet velocity did not match printing speed, and that the inlet velocity was greater than its optimal value. As shown in Figure 9b, when the inlet velocity was 0.0343 mm/s there was a slight accumulation in the printing model, indicating an improvement in the mismatch degree between the inlet velocity and the printing speed, with the inlet velocity slightly higher than the optimal value. As shown in Figure 9c, when the inlet velocity was 0.0250 mm/s, the forming quality was higher and the model was smooth and flawless, indicating that the inlet velocity matched the printing speed at this time and was close to its optimal value. As shown in Figure 9d, when the inlet velocity was 0.0156 mm/s, the extrusion wire was broken, indicating that the inlet velocity did not match the printing speed and was less than its optimal value. In summary, printing quality could be guaranteed at the inlet velocity close to its optimal value at 0.0242 mm/s, indicating that the matching between inlet velocity and printing speed was an interval value. In this article, we study the relationship between extrusion speed and deposition line width by setting the printing speed. This is consistent with the research ideas of Cao et al. [31], who used a constant extrusion speed to study the relationship between printing speed and line stretching rate. The results indicate that when the printing speed remains constant within a certain range, the actual line width gradually increases with the increase of extrusion speed.

### 3.4. Effect of Initial Gelatin Solution Temperature on Temperature Field Distribution

During extrusion, the gelatin solution first transferred some heat to the printing head, and then exchanged heat with cold air. If the initial temperature of the gelatin solution was too high, it would take a longer time for the gelatin solution to cool down to the gel point. The gelatin solution failed to solidify rapidly after deposition, resulting in a decrease in the printing accuracy. Therefore, the initial temperature of the gelatin solution was one of the important factors affecting printing quality.

Guo et al. found that the gelatin gel process can be divided into two stages: the fast gel stage and the slow gel stage. When the temperature of 9 g/dL gelatin solution is lower than 31 °C, gelatin enters into the slow gel stage, and when the temperature is lower than 26 °C, gelatin enters into the fast gel stage [32]. Zhao et al. studied the influence of temperature on the precision of gelatin molding. The results showed that a 10 w% gelatin solution was in the form of liquid drop, pre-gelation, gelation, and super-gelation at 27, 26, 25, and 24 °C, respectively, and the lines were more uniform when it was close to gelation [15].

This means that under ideal printing conditions the gelatin solution should be at a rapid set point when extruded into the print nozzle outlet, allowing the gelatin solution to set quickly after deposition. However, in practice there is a certain difference between the temperature of the added gelatin solution and the temperature of the extruded gelatin solution, which requires consideration of the heat exchange between the gelatin solution and the barrel wall and print head during the extrusion process. So, in this section the variation of the temperature of the material in the exit section of the gelatin aqueous solution at different initial temperatures over time and the effect on print quality is explored. Figure 10 shows the change curve of gel temperature with time under the conditions of inlet velocity of 0.03 mm/s, ambient temperature of 5 °C, nozzle diameter of 1 mm and heating jacket temperature of 45 °C. As shown in Figure 10, with the initial temperature of 35 °C and 30 °C, the temperature at the exit section of the gelatin solution decreased gradually and then became stable with the increase in time. At 35 °C, the temperature of the gelatin solution decreased slowly and it took a long time to reach the equilibrium temperature. At 3 °C, the temperature of the gelatin solution quickly stabilized, getting close to the rapid setting temperature. At 25 °C, the temperature of the gelatin solution firstly dropped, then rose, and finally tended to reach the equilibrium. At 25 °C, the lowest temperature of the gelatin solution in the outlet section was 21 °C. Therefore, under the above process conditions, the optimal initial temperature of the material should be about 30 °C.

Figure 11a shows the printing results of the gelatin solution at an initial temperature of 35 °C, which indicates that the model exhibited stacking deformation. Figure 11c shows the printing results of the gelatin solution at an initial temperature of 25 °C, accompanied by particle-like extrusion lines. Figure 11b shows the printed results of the gelatin solution at an initial temperature of 30 °C. The model has no flow or deformation phenomena, and the molding quality is high, which is consistent with the simulation results.

### 3.5. Effect of Material Concentration on Flow Field Distribution

At low shear rates, viscosity was related to the fluidity of the material before extrusion [8]. Therefore, it was very important to study the influence of viscosity variation with numerical values on other parameters such as shear rate and pressure field. As shown in Figure 12a,b, the distribution of velocity and shear rate fields in the flow channel did not change with material viscosity, mainly depending on inlet velocity. As shown in Figure 12c, the pressure inside the barrel remained basically constant in the Z-axis direction, and the higher the concentration, the greater the pressure inside the barrel. The pressure gradually decreased from the barrel to the outlet, and the pressure of the gelatin solution decreased more rapidly with the increase in concentration, while the pressure at the nozzle outlet basically remained unchanged. These results are comparable to the research results of Yang et al. [5].

In the actual 3D printing process, the extrusion process was not instantaneously extruded to the viscosity of colloidal materials. Only when the extrusion pressure was greater than the material viscosity could it be smoothly extruded. The extrusion response time was affected by material viscosity, and the higher the viscosity, the greater the pressure required for material extrusion. The longer the extrusion response time, the greater the hysteresis, which was not conducive to printing. Therefore, pressure in the printing process could be ensured within a reasonable range by adjusting the concentration of the gelatin solution, and the material viscosity could be adjusted to an appropriate value.

The printing results of gelatin solutions with concentrations of 12%, 9%, 6%, and 3% were obtained at an ambient temperature of 5 °C, an initial temperature of 30 °C, and an inlet velocity of 0.025 mm/s, as shown in Figure 13. The storage modulus (G′) is a measure of elastic solid materials, which reflects the mechanical strength of the specimen. Materials with high mechanical strength exhibit excellent self-supporting properties after deposition and maintain a stable shape after printing.

As shown in Figure 13a,b, the printing accuracy of 12% and 9% gelatin solutions is relatively high. As shown in Figure 13c, 6% gelatin solution had insufficient support performance, resulting in the serious deformation of the printing model. As shown in Figure 13d, 3% gelatin solution flowed obviously during printing, making it difficult to form. Therefore, the energy storage modulus of gelatin was proportional to the concentration of gelatin, and when the gelatin concentration was below 6% its self-supporting ability was far from being sufficient for formation. These results are comparable to the research results of He et al., who showed that in the range of 10~20% MA modified gelatin gel, the higher the concentration, the greater the Young’s modulus, and the better the mechanical properties [33].

### 3.6. Effect of Nozzle Diameter on Flow Field Distribution

The nozzle diameter was a key parameter affecting printing time and accuracy. Using the nozzle diameter as a variable, we explored its impact on the parameters inside the flow channel. As shown in Figure 14a,b, there was no significant change in the velocity and shear rate in the flow channel of the printing head with different diameters in the barrel section, but they rapidly increased in the printing head section. The smaller the printing head diameter, the more rapidly the velocity and shear rate increased.

As shown in Figure 14c, as the nozzle diameter increased from 0.5 mm to 2.0 mm, the relative pressure inside the material barrel decreased from 19.9 Pa to 3.3 Pa, and the pressure declined significantly by about 83.5%, which indicates that the smaller the nozzle diameter, the greater the pressure required for material extrusion and the more high-pressure zones there are in the flow channel. The pressure difference between the inlet and outlet of the printing heads with diameters of 0.5 mm, 1.0 mm, 1.5 mm, and 2.0 mm was 19.9 Pa, 7.8 Pa, 4.6 Pa, and 3.3 Pa, respectively. The above results demonstrated that the smaller the nozzle diameter, the greater the pressure difference between the inlet and outlet. However, the larger the pressure difference, the more elastic potential energy is released after extrusion, which is very likely to result in expanded extrusion. Therefore, the nozzle with an appropriate diameter should be selected in the actual printing process. These results are comparable to the research results of Oyinloye et al. [34].

## 4. Conclusions

Simulation techniques can be used to predict the distribution of the flow field in the flow channel during the printing process, as well as the influence of process parameters on the flow field distribution. It was found that the parameter changes within the barrel during the low-temperature extrusion process were small, and the parameter changes were mainly concentrated in the nozzle section. As the inlet velocity increases, the velocity and shear rate in the flow field increase, and the material temperature in the outlet section increases. The material concentration and temperature are the main factors affecting the viscosity; as the gelatin concentration increases, its viscosity, storage modulus, and self-supporting ability increase. As the print head diameter decreases, the pressure, velocity, and shear rate in the flow channel increase.

The initial temperature of gelatin mainly affects the distribution of the temperature field in the flow channel and the temperature of the gelatin aqueous solution in the outlet section. In an ideal printing state, when the gelatin solution is extruded to the outlet of the printing nozzle, the temperature of the gelatin solution should be at the rapid condensation point temperature. When the temperature of the gelatin aqueous solution in the outlet section is lower than the rapid condensation point temperature, it can easily cause the gelatin aqueous solution to solidify in the printing head and cause extrusion blockage. When the temperature of the gelatin in the outlet section is greater than the rapid setting temperature of the gelatin, it can easily cause the print model to flow and collapse.

Through experiments and simulations, it was found that for the experimental 3D printer—under the process conditions of: environmental temperature, 5 °C; heating jacket temperature, 45 °C; nozzle diameter, 1 mm; and printing speed 5 mm/s—the molding quality was better when the inlet speed was 0.025 mm/s, the initial temperature of the gelatin solution was 30 °C, and the gelatin concentration was 9% and 12%. When the concentration of gelatin solution was below 6%, it was difficult to form due to insufficient self-supporting ability. The process parameters obtained from the experiment were within a range, which could also meet the molding process requirements when the inlet speed and the initial temperature of the gelatin solution were close to the experimental conclusion. This paper provides a certain reference value for the optimization of the low-temperature deposition 3D printing process for other food materials and the design of a low-temperature deposition 3D printer extrusion mechanism. Materials are an important factor limiting the development of 3D printing at present. In the future, the printability of more kinds of hydrogels can be explored, providing more possibilities for food and biological 3D printing.

## Figures and Tables

**Figure 1 foods-12-02687-f001:**
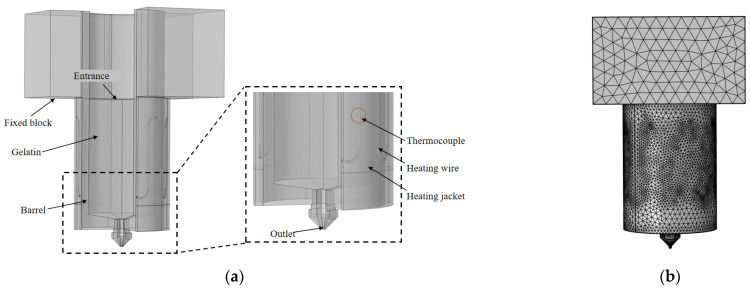
(**a**) 3D model of extrusion device; (**b**) Grid diagram of extrusion device.

**Figure 2 foods-12-02687-f002:**
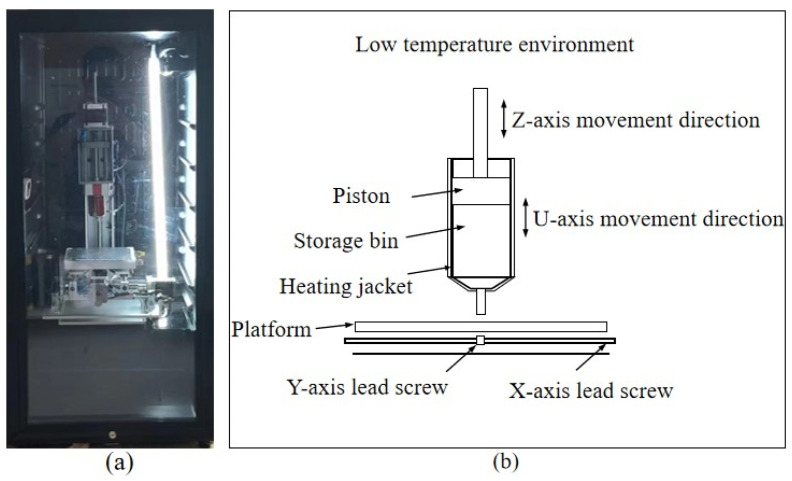
Schematic diagram of the structure of the cryogenic deposition 3D printer: (**a**) Equipment physical diagram; (**b**) Equipment diagram.

**Figure 3 foods-12-02687-f003:**
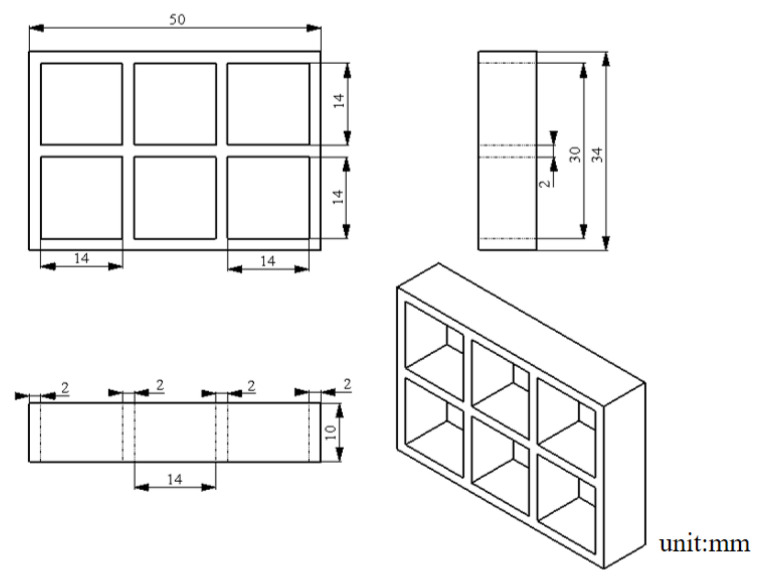
Schematic diagram of grid structure.

**Figure 4 foods-12-02687-f004:**
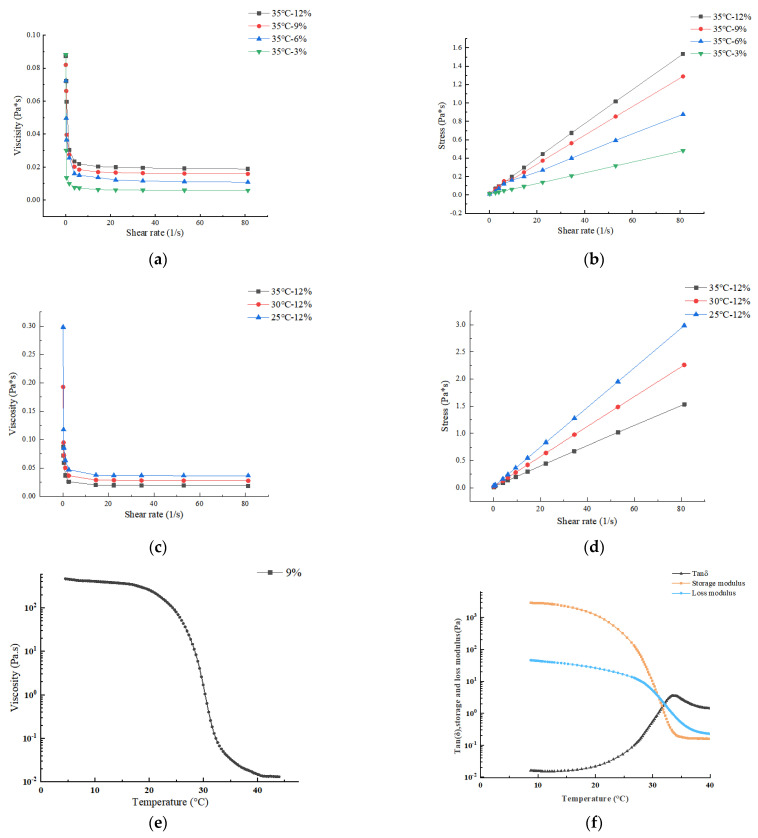
Material rheological properties: (**a**) Apparent viscosity versus shear rate curve of different concentrations of gelatin solution; (**b**) Stress versus shear rate curve of different concentrations of gelatin solution; (**c**) Apparent viscosity versus shear rate curve of gelatin solution at different temperatures; (**d**) Stress versus shear rate curve of gelatin solution at different temperatures; (**e**) Apparent viscosity versus temperature curve of 9% gelatin solution; (**f**) Temperature-dependent curves of storage modulus and loss modulus of 9% gelatin solution.

**Figure 5 foods-12-02687-f005:**
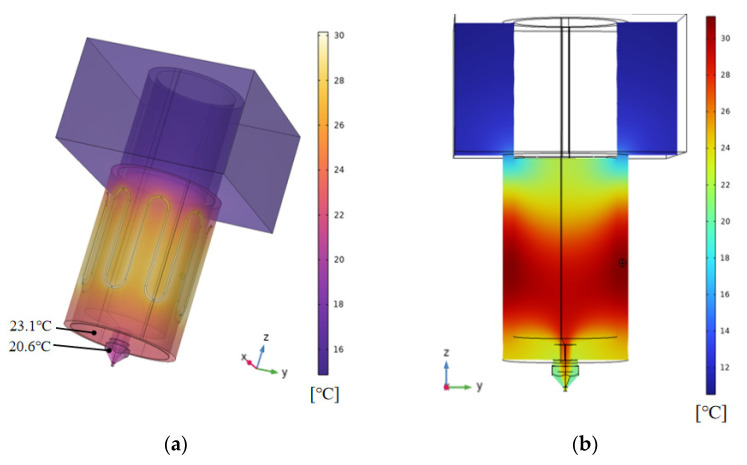
Simulation of channel flow field distribution based on Comsol: (**a**) Three-dimensional temperature distribution; (**b**) Channel temperature distribution; (**c**) Channel velocity vertical distribution; (**d**) Outlet section velocity horizontal distribution; (**e**) Channel shear rate vertical distribution; (**f**) Outlet section shear rate horizontal distribution; (**g**) Channel viscosity vertical distribution; (**h**) Outlet section viscosity horizontal distribution; (**i**) Channel pressure vertical distribution; (**j**) Outlet section pressure horizontal distribution.

**Figure 6 foods-12-02687-f006:**
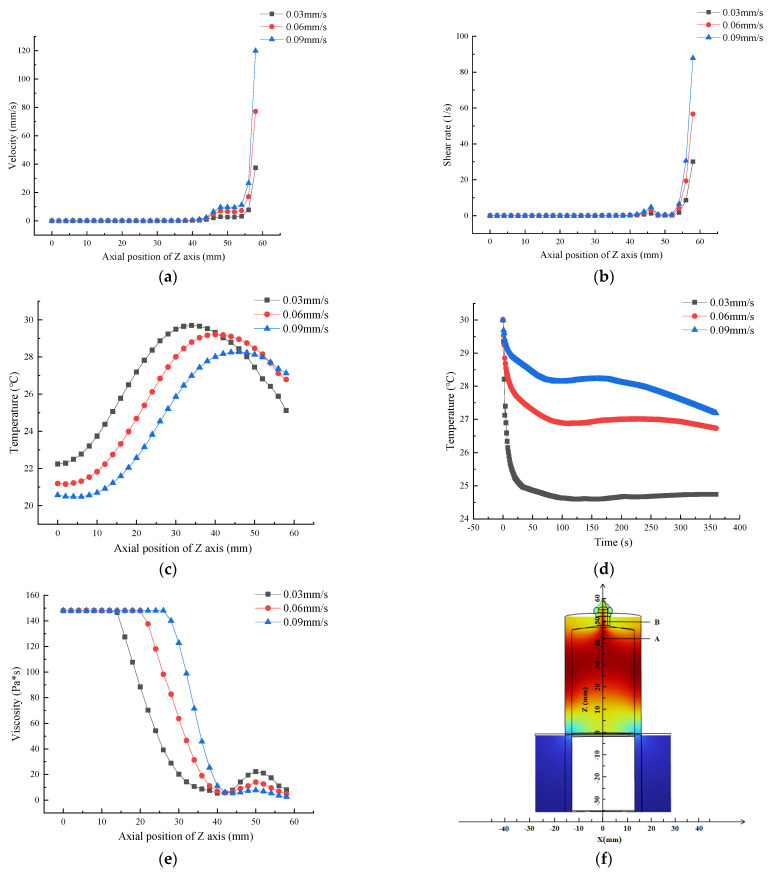
Effect of inlet velocity on flow field distribution based on Comsol simulation: (**a**) Velocity; (**b**) Shear rate; (**c**) Temperature; (**d**) Temperature variation curve of material in the outlet section with time; (**e**) Viscosity; (**f**) Coordinate distribution.

**Figure 7 foods-12-02687-f007:**
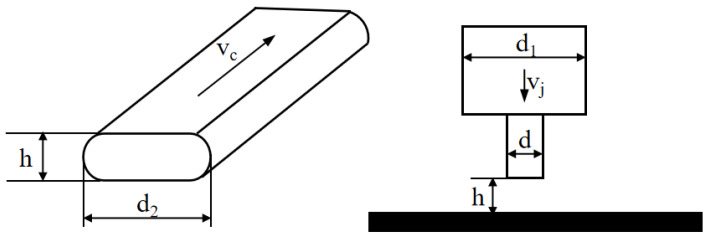
Line model.

**Figure 8 foods-12-02687-f008:**
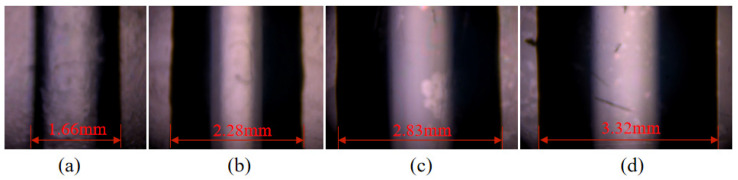
The effect of inlet velocity on the deposition width of wire: (**a**) 0.0156 mm/s; (**b**) 0.0313 mm/s (**c**) 0.0469 mm/s; (**d**) 0.0625 mm/s.

**Figure 9 foods-12-02687-f009:**
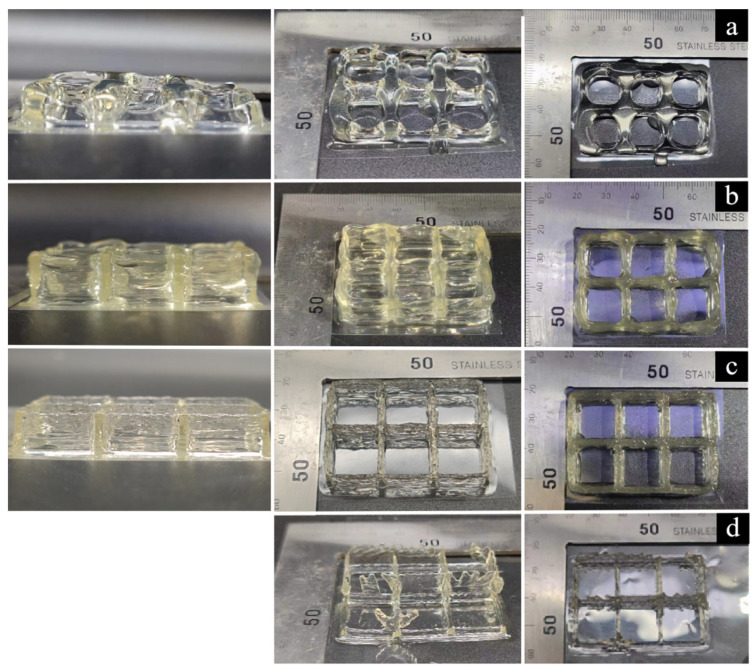
The effect of extrusion speed on 3D printing quality: (**a**) 0.0438 mm/s; (**b**) 0.0343 mm/s; (**c**) 0.0250 mm/s; (**d**) 0.0156 mm/s.

**Figure 10 foods-12-02687-f010:**
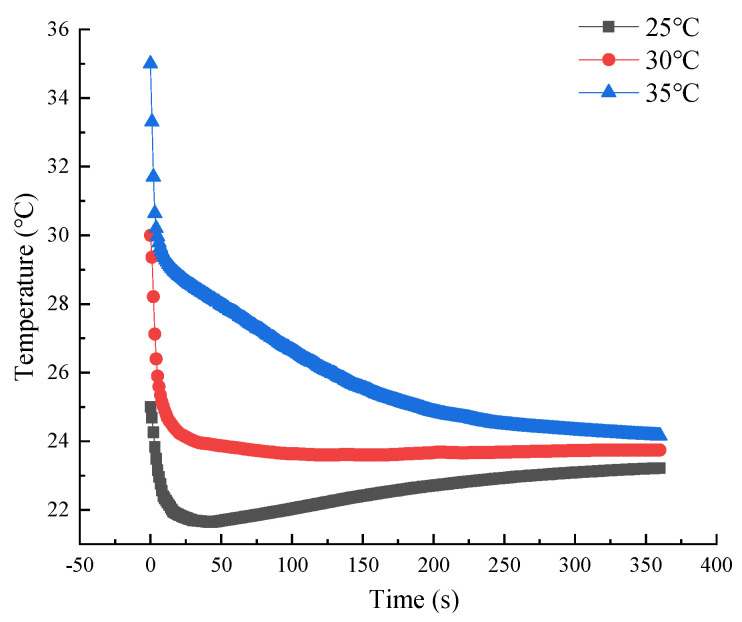
Temperature versus time curve of gelatin solution exit section material at different initial temperatures based on Comsol simulations.

**Figure 11 foods-12-02687-f011:**
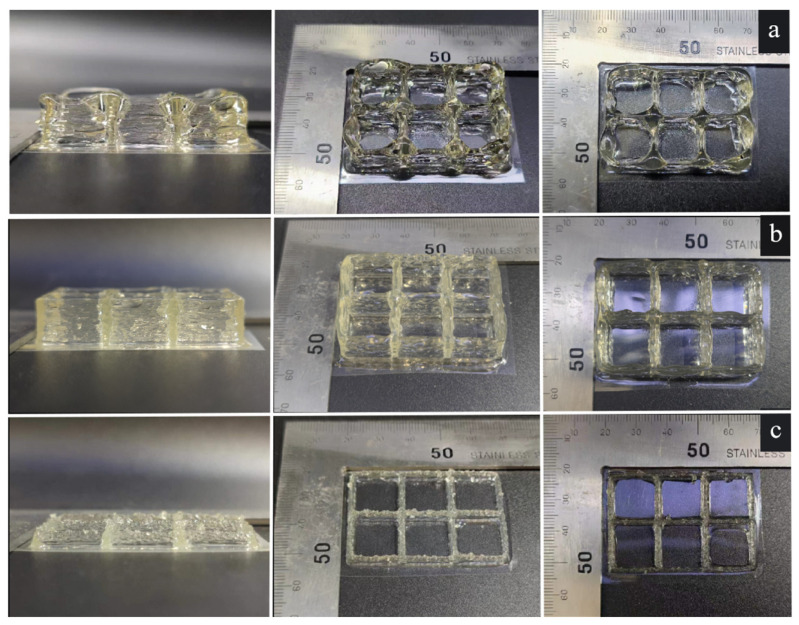
Gelatin printing effect at different initial temperatures: (**a**) 35 °C; (**b**) 30 °C; (**c**) 25 °C.

**Figure 12 foods-12-02687-f012:**
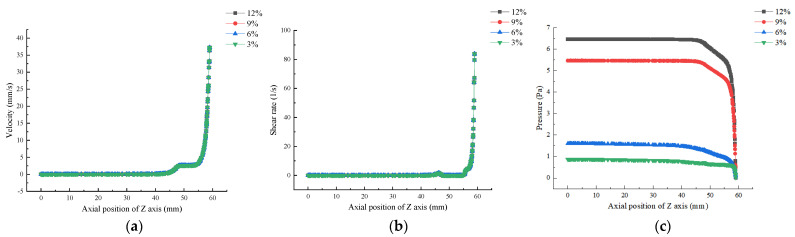
Effect of material concentration on flow field distribution based on Comsol simulation: (**a**) Velocity; (**b**) Shear Rate; (**c**) Pressure.

**Figure 13 foods-12-02687-f013:**
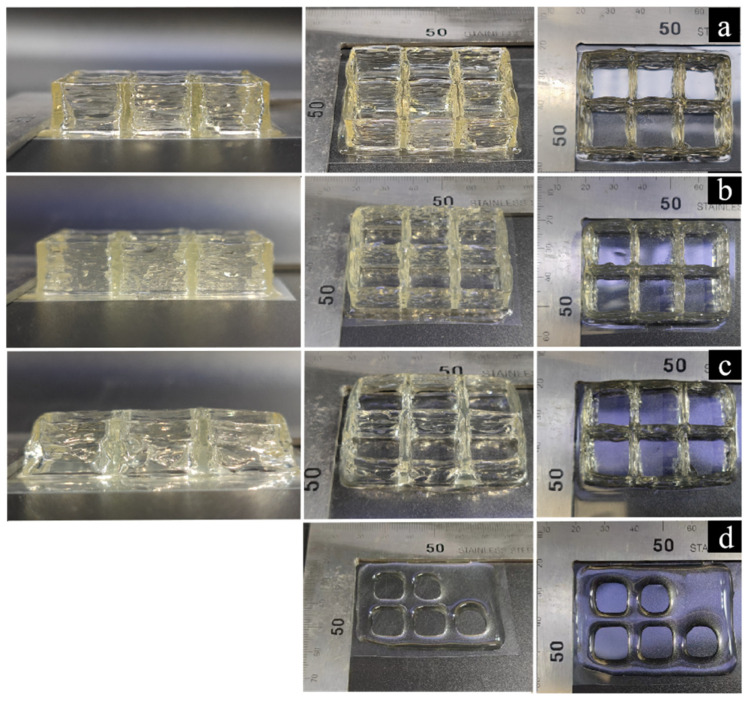
The effect of material concentration on 3D printing quality: (**a**) 12%; (**b**) 9%; (**c**) 6%; (**d**) 3%.

**Figure 14 foods-12-02687-f014:**
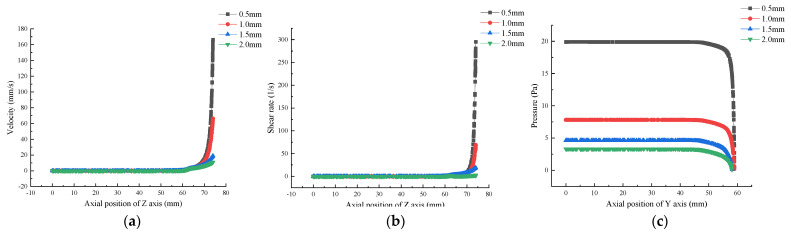
Effect of nozzle diameter on flow field distribution based on Comsol simulation: (**a**) Velocity; (**b**) Shear rate; (**c**) Pressure.

## Data Availability

The datasets generated for this study are available on request to the corresponding author.

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
