# Peer review of "A Study of the Gelatin Low-Temperature Deposition Manufacturing Forming Process Based on Fluid Numerical Simulation"

_foods, 2023, doi:10.3390/foods12142687_

Round 1
Reviewer 1 Report
Paper merits publication in Food. It is well structured, and gives essenatial information of practical meaning in 3D printing techique utilising geltian. Paper can be useful for readres involved in this technology.
Proior the publication it is suggested to check spelling ( line 254 - omitted letter S) .
I am not sure if among other conclusions, the statement "The viscosity of material decreases with the increase of temperature", is necessary - this is typicla property of gelatin therefore it is proposed to eliminate it.
Reviewer 2 Report
For Figure 1: What s the model for? Please write the sentence as ‘Figure 1. Physical model of what
What s the reference of Measurement of rheological properties?
A graphical abstract can be prepared and put in the manuscript.
What is the reason for selecting 12 wt%, 9 wt%, 6 wt%, and 3 wt% gelatin aqueous solution?
Generally, spaces between the (24.3 ℃) number and ℃ should be deleted in all manuscript document.
The title of 3.4. hear rate distribution should be rearranged as ‘ shear rate distribution’.
Resolutions of the figures should be checked because they can not be read.
The future of the perspective should be improved in the discussion section.
All the results should be compared/supported with the references. The comparing results with the literature can be improved.
Round 2
Reviewer 2 Report
The revised version was formed.